# Simulating and Contrasting the Game of Open Access in Diverse Cultural Contexts: A Social Simulation Model

Oswaldo Terán [1,2] and Jacinto Dávila [2,*]

1   Escuela de Ciencias Empresariales, Universidad Católica del Norte, Coquimbo 1780000, Chile; oswaldo.teran@ucn.cl
2   Centro de Simulación y Modelos (CESIMO), Universidad de Los Andes, Mérida 5101, Venezuela
*   Correspondence: jacinto.davila@gmail.com

**Abstract:** Open Access is a global cause with the aim of allowing unrestricted access to all scientific research output in electronic formats. This paper presents a model for simulating the game of interests behind this cause in order to investigate ways of promoting the practice of open access. The model represents the following actors: Academics, Administrators, Funders, Publishers and Politicians. Five scenarios were developed to represent both realistic and ideal, interesting, situations. The model was developed using the SocLab platform—a formalization of the sociology of organizational action. It is based on previous descriptions of the game and expert knowledge. A structural analysis permits us to examine the properties of the sub-model behind each scenario. The results corroborate certain intuitions about the scenarios representing realistic cases, e.g., they indicate that publishers, being isolated in their interests, are subject to strong pressures from other actors, who have a circumstantial alliance. Administrators take an intermediate stance in all scenarios. The best scenarios for open access are those in which Politicians and Funders clearly support the cause by expressing mandates in that direction, backing academics. Surprisingly, the model shows that it is in the Publishers' interest not to take an extremist position against open access.

**Keywords:** open access; scientific research; game of interests; sociology of organizational action; social simulation; SocLab





## 1. Introduction

Open Access, as defined by the Budapest Open Access Initiative [1], is a global initiative that strives to provide unrestricted access to all research outputs from the scientific community in electronic form. This vision is based

> *"on the principle that making research freely available to the public supports a greater global exchange of knowledge"*

that numerous research communities, all around the world, have explicitly embraced (e.g., [2–19]). However, conflict arises among those who believe in some form of that principle and those who believe that the publishing status quo is a better solution. This conflict is primarily economic in nature, as supporting open access entails removing access charges for content distribution, which typically involves certain costs that need to be covered. The publishing status quo promotes access charges, which is called Toll or Readers' Subscription Access (TA).

Scientific publishing, however, has been influenced by economic concerns since its very early history. As explained in [20], the origins of the concept of intellectual property can be traced back to a (very successful) attempt to extend the concept of landed property: "This bit of legal creativity was actually motivated by the stationers who needed to establish legally viable claims over the texts they printed, if only to protect their trade from imitation and piracy. To them, this meant exclusive and perpetual ownership, as is the case for land property. But they were not the only players [or actors] and, as a result of various court

actions, the definition of what they actually claimed to own remained murky for several decades, almost a century, actually" (ibid, Chapter 3).

With the arrival of the Internet and 21st-century electronic technology, making a copy of a book became a process with almost zero cost (apart from the costs of producing the book in the first place). In this context, the actual value of the text (images included) can be almost completely mapped to its content and eventually to its authors. This new situation has triggered a wave of questions about the origins of that value and who is entitled to use and enjoy each contribution. This line of questioning is particularly striking in scientific practice, where scientists constantly seek to access ideas and proposals by other scientists while trying to solve scientific problems.

The effects of transitioning from ink-on-paper to digital or electronic texts were outlined in [21]. This work defines the ideal conditions for the "grand conversation of science", where authors are free to read and share their ideas during the process of designing, establishing, reporting, and refuting theories and experiments. These conditions are known as Open Access (OA), and can be summarized as follows: "Open access means that scientific literature should be publicly available, free of charge on the Internet so that those who are interested can read, download, copy, distribute, print, search, refer to and, in any other conceivable legal way, use full texts without encountering any financial, legal or technical barriers other than those associated with Internet access itself" (Declaration of Berlin [22]).

It is important to clarify that OA is made possible by the Internet and the consent of copyright holders, as highlighted by Suber ([21], p. 9): "OA isn't an attempt to bypass peer review [...]. OA isn't an attempt to reform, violate, or abolish copyright [...]. OA isn't an attempt to deprive royalty-earning authors of income [...]. OA isn't an attempt to deny the reality of costs [...]. OA isn't an attempt to reduce authors rights over their work [...]. OA isn't an attempt to reduce academic freedom [...]. OA isn't an attempt to relax rules against plagiarism [...]. OA isn't an attempt to punish or undermine conventional publishers [...]. OA doesn't require boycotting any kind of literature or publisher [...]. OA isn't primarily about bringing access to lay readers [...]. Finally, OA isn't universal access [...]" [21], pp. 20–27.

At this juncture, it must be evident that divergent interests and significant confusion surround this topic [23] and that interactions among the involved parties are governed by these interests. To gain a better understanding of the dynamics among these actors, it is crucial to study the system through modeling and social simulation. The primary motivation for this study is two-fold: firstly, to comprehend the interactions among actors in this system; and secondly, to explore strategies and actions that can promote the practice of open access. However, it should be noted that not all actors in the system share the same level of interest in this research question; for example, it is of interest to academics but not necessarily to publishers.

To achieve this, we developed a social simulation model within the SocLab framework [24–26] that provides a formalization of the Sociology of Organizational Action [27]. The model, along with its data modules, can be accessed at [28] for those interested in replicating our results or testing alternative scenarios. In this game, a desirable situation for open access is characterized by academics being satisfied and a stable conflict where satisfaction and power are reasonably distributed among the actors (these variables will be defined later). We developed several scenarios representing different interpretations of the underlying interests and conditions in academic publishing, drawing from working experience and activism of domain experts. The model in each scenario serves as a tool to understand the connections between actors' motivations and their simulated behavior, which can then shed light on actual behavior in the real world. The model serves to enhance understanding and corroborate intuitions regarding realistic situations.

Social simulation involves modeling social systems, including economics, organization, politics, history, and social–ecological systems, and studying their behavior and emergent properties through computer simulations [29–31]. The Sociology of Organized Action investigates how social organizations become regularized through counterbalancing



processes that involve power relationships among social actors. SocLab allows for the simulation of social games, where actors adjust their behavior in relation to others to achieve a satisfactory level of capability in reaching their goals, which combines the aims assigned to their roles within the organization and their individual objectives. In addition to simulating the social game, SocLab facilitates the study of the model's structural properties through analytical tools and exploration of the space of possible configurations, including Pareto optima, Nash equilibrium, structural conflicts, and more. The SocLab framework has been applied in the study of concrete organizations based on sociological inquiries [24–26].

In a SocLab game, the model's construction involves identifying the actors and the resources they control. The input data encompass the actors' interests in the resources, the effects of resource states on actors' behavior, and the solidarities between actors. The game's initial state is determined by the state of the resources, and at each simulation time step or iteration, actors play the game by adjusting the state of the resources they control until a stable state is achieved. Actors modify the levels of their resources to increase their satisfaction, and when all actors surpass their aspiration levels of satisfaction, the simulation run stops, and this configuration is referred to as the regulated state. Actors make decisions based on predefined rules of action that evolve during the game, and they also have aspiration levels that decrease as the game progresses. Solidarities among actors are introduced to model how they account for each other, where negative values represent hostility, zero indicates a lack of concern, and positive values represent realistic solidarity. Satisfaction and influence are mediated by the solidarities between actors, with an actor's satisfaction increasing when they and those they have positive solidarity with receive positive capacity, and decreasing when those they have negative solidarity with receive positive capacity. Similarly, an actor's influence is affected by the solidarity they exhibit towards others.

A SocLab game offers various aspects that can be analyzed, such as the distribution of satisfaction and influence in the regulated state. By studying the structure of the game, configurations that differ from the regulated state, such as maximum satisfaction of an actor or Nash equilibrium, can be examined. The structural analysis considers all possible configurations of the SocLab game, not solely those of the regulated state.

The remainder of this paper is organized as follows. Section 2 presents a realistic description of the working case, namely, the open-access game in science (especially in Europe and North of America). Then, Section 3 describes the model of the case in SocLab: the open-access SocLab model. Subsequently, Section 4 presents the results obtained from simulating the case and various scenarios, as well as an analysis of the structure of the model. Finally, sections five and six offer a discussion and conclusions, respectively.

## 2. Open Access in Science as a Game of Interests

Open Access as a game [32] has been considered before [33]. In that study, the authors identified several key actors involved in the game of scientific knowledge and open access. These actors included: 1. Researchers; 2. Graduate students; 3. Junior faculty; 4. Senior faculty; 5. Librarians; 6. Administrators; 7. Funders of research; 8. Publishers; and 9. Politicians.

These actors represent different groups of people who play various roles in the process of scientific knowledge generation and distribution. The authors acknowledge that these categories represent a reduction in complexity, as real-world actors have many other characteristics and a distributed worldview.

Moreover, it is important to note that the communities involved in scientific research have dynamics and factors that are not explicitly captured in this modeling exercise. One such important dynamic is the assessment and evaluation of researchers and scientific institutions. When researchers choose a journal to which to submit their work, their decision is often influenced by their intention to obtain a better assessment or evaluation. Metrics such as the impact factor of a journal have traditionally been used as proxies for determining the quality and prestige of a publication. It is crucial to acknowledge that this

assessment process is highly complex and multifaceted. The dynamics of journal selection and publication choices are influenced by a variety of factors, including the researchers' career goals, disciplinary norms, institutional expectations, and the perceived reputation of journals. The use of impact factors as a measure of quality has also been subject to criticism and debate within the scientific community. In this modeling exercise, we made the assumption that open-access journals also publish papers of good quality.

To further simplify the modeling process, the authors aggregated some actors based on their interests and goals regarding publishing. They grouped together the following actors: Group 1: Researchers, Graduate students, Junior faculty, Senior faculty, and Editors who donate their editorial work to publishers (referred to as "Academics"); Group 2: Librarians and Administrators (referred to as "Administrators").

This aggregation allowed the authors to reduce the number of distinct actors in the game while still considering the main interests and goals of the various groups related to publishing. The action possibilities of these aggregated groups are provided in Table 1.

**Table 1.** Actors and their actions. Second approximation.

| Actors | Actions |
|---|---|
| Academics | Publish Toll Access, TA<br>Publish Open Access, OA<br>Perish |
| Administrators | Support TA<br>Support OA<br>Support Both |
| Funders | Demand publications of any kind<br>Demand OA publications<br>Demand nothing |
| Commercial Publishers (Editorial houses) | Grant TA<br>Grant OA<br>Grant big deals<br>Grant OA with embargoes |
| Politicians | Permit TA<br>Demand Green OA<br>Demand Gold OA<br>Demand Some OA |

By using the SocLab social simulation tool, here we aimed to model the interactions among these actors and understand how open access as a game might unfold, considering the interests and actions of the different groups involved. The simulation process could provide insights into the dynamics of the academic publishing system and shed light on potential scenarios for promoting open access.

There are, of course, many subtleties in the reduction presented in Table 1 that are not captured by distinguishing different types of actors and their actions. The actual role that each actor plays in the game will have to be modeled by the preferences it expresses when it comes to act, that is, by choosing one action and not others. For instance, researchers may have different opinions about open access, some prefer toll access for their own reasons, and some may even be misinformed. Therefore, the model must allow for some support for each action type within the same actor type. We believe that an increasing number of researchers will realize that publishing their own papers for a fee while accessing others' papers for free is not a fair game (the fairest being one where everyone is free to read any contribution without a toll). Moreover, visibility, prestige, and promotion can also come from open-access journals, and article processing costs tend to be the same as better automatic tools become available. However, not everybody is equally informed of the bigger picture. Therefore, in this work the models allowed for different levels of support for each action to take alternative views into consideration. In [33], the researchers were

initially tempted to simplify a similar model by assuming that Funders and Politicians are the same group of people. However, they realized that this simplification would result in an under-representation of the important effects and outcomes in the game. They recognized the need to balance the reduction in the model by acknowledging the absence of an explicit representation of society as a whole, which plays a crucial role as the final receiver, consumer, and potential user of the knowledge generated by the academic ecosystem.

To address this, they decided to retain the Politician actor and model the expected outcomes as discrete fields of selected variables, as outlined in Table 2. The chosen variables are associated with the actions of each actor group and contribute to determining the utility for each actor. Instead of consolidating a mathematical expression of utility, the researchers assigned a finite set of possible values to each variable, creating a discrete universe of possibilities to explore.

**Table 2.** Actors, actions and outcomes (with the sets of possible values).

| Actors | Actions | Outcomes {Possible Values} |
|---|---|---|
| Academics | Publish TA<br>Publish OA | Opportunity {Maximal, Minimal}<br>Visibility {More, Less}<br>Prestige {More, Less}<br>Promotion {More, Less}<br>Savings for society {More, Less}<br>Quality results {More, Less}<br>Income for publishers {More, Less}<br>Societal impact & relevance {More, Less} |
| Administrators | Support TA<br>Support OA | |
| Funders | Demand TA publications<br>Demand OA publications | |
| Publishers (Editorial houses) | Grant TA<br>Grant OA | |
| Politicians | Permit TA<br>Demand OA | |

The researchers in [33] further simplified the game and explored ways to "resolve" it. They arrived at a highly simplified form of the game with only two actors, where the strategy profile of Academics choosing Open Access and Publishers opting for toll access represented a Nash equilibrium. This finding suggested that Publishers would continue to prioritize toll access models even if the entire academic community embraced Open Access. However, this conclusion was reached in a two-actor game, and while this allowed easier deductions and calculations, it may not be readily extrapolatable to realistic situations.

Nonetheless, the selection of actors in the model does support the modeling of "the tensions between [...] the twin objectives -- of money and mission" that have characterized scientific publishing for a significant period of time [34]. The behavior of one of these actors in the model reflects aggregate performances as the expected behavior of a whole community. The corresponding real community does not have to be homogeneous, and so might have individuals and sub-groups with diverse, even opposite, behavior, which will indeed be represented. In the following sections, this paper describes the basis of the new modeling tool used in this study, which was informed and enriched by the previous work discussed in this section.

## 3. The Game of Open Access in SocLab: Scenario 0

We suggest that the reader examine Appendices A and B before reading this section. It gives a formal explanation of the game implemented on the SocLab platform, i.e., the simulation methodology. It describes how SocLab operates, defining the main variables and constructs, e.g., on the one hand, model inputs such as solidarities among actors, actors' stakes in resources, and the effect functions of resources on actors, and, on the other side, outputs variables such as states of the resources and actors' influence and satisfaction.

We intend to describe the regional peculiarities of how the game is played in different places that have justified a set of different scenarios explored in the simulations. It can be said that the open access game started in Europe among academic institutions, on

the continent where the status quo of international academic publishing was initially questioned (e.g., the Budapest Open Access Initiative [1] and the Declaration of Berlin [22]). It has spread throughout the continent towards other places with closer ties to Western science. Therefore, it is no surprise that countries such as Germany, France, the Netherlands, the United Kingdom, Canada, and the United States of America display trends toward common collective behavior, and there have even been some landmarks victories for activism. Among these landmarks, national and even continental political mandates have been issued for the research community [35]. However, non-governmental organizations with great influence as funders for scientific research have declared themselves in favor of Open Access, such as the Wellcome Trust [36] in the UK and the Bill and Melinda Gates Foundation [37] in the USA. This comes following a long tradition of public open access in medical disciplines in the USA (NFS [38]). These trends are represented by Scenario 0. The potential behaviors according to the state of the resources they control are described in Table 3.

**Table 3.** Actors' behavior in accordance with the state of the resources they control.

| Actor–Resource | Actor's Behavior |
| --- | --- |
| Academics–PublishOA | From disfavoring (Low) to clearly favoring (High) the publishing of their contributions (papers) as open-access documents |
| Administrators–SupportOA | From ignoring (Low) open-access journals to including and encouraging (High) the use of those journals for academic publications in their institutions |
| Funders–DemandOAPublications | From ignoring open-access publications (maybe all publications) (Low) to demanding open access to the research resulting from projects they fund (High) |
| Publishers–GrantOA | From not publishing anything as open access (Low) to only publishing as open access (High) |
| Politicians–DemandOA | From ignoring open-access publications (maybe all publications) (Low) to demanding open access to the research resulting from projects they fund (High) |

Other regions with different cultural backgrounds and legal, economic, and scientific particularities, such as Latin America, show different behaviors that, from the perspective of domain experts, can be better modeled by Scenario 1 (there are few well-funded initiatives promoting the open access existing in Latin America, but among them is SciELO (Scientific Electronic Library Online). For some local actors who are interested in open access in these regions, Scenario 0 may represent a better or ideal situation. The gap between Scenarios 0 and 1 represents the problem that they may wish to solve. The data and analysis below may shed light on how to proceed.

While the two cases represented by Scenarios 0 and 1 show common and realistic forms of the game, other realistic situations could be seen as a mix of these two. Additional situations of interest for better understanding the games and their limitations are represented by Scenarios 2–4. These scenarios are based on variations of Scenario 0 (in this sense, Scenario 0 is considered as the base model). Scenarios 2 and 3 investigate how a more active position of Publishers with respect to their own benefit affects the game and open access. Publishers are less permissive than in Scenario 0, but to different degrees, being much less permissive in Scenario 3. Scenario 4 examines how the higher awareness on the part of Funders of the importance of open access compared to Scenario 0 promotes open access. To this end, Funders direct an considerable portion of their positive solidarity towards

Academics. Below, more detail about Scenario 0 is presented, and all of the scenarios are better explained in Appendix B.

The model of each scenario is a useful tool for understanding the connections between what we interpret as the basic motivations of each actor and their simulated behavior, which could be further used to explain their actual behavior in the world. The validation of these behaviors, however, is beyond the scope of this work.

### 3.1. Scenario 0

The game for Scenario 0 is presented in Tables 4 and A1 of Appendix B. We will use the name of the type of actor, e.g., Academics, to refer to the actors in the model. The model consists of the five actors mentioned previously and five resources, which represent the types of actions each actor is responsible for based on the actions described previously. In this case, however, a resource indicates a policy for action that covers a continuum from acting completely against to fully supporting open access, with the specific actions of each actor. The way of describing the game elaborated here permits the definition of functions for each resource, as well as relating the actions of each actor with their effects.

**Table 4.** Table of solidarities among actors. Each cell shows the (intensity of the) solidarity that the actor in the row expresses towards the actor in the column.

| Solidarity → | Academics | Administrators | Funders | Publishers | Politicians |
|---|---|---|---|---|---|
| Academics | 1.0 | 0.0 | 0.0 | 0.0 | 0.0 |
| Administrators | 0.0 | 0.9 | 0.0 | 0.1 | 0.0 |
| Funders | 0.0 | 0.0 | 1.0 | 0.0 | 0.0 |
| Publishers | 0.0 | 0.0 | 0.0 | 1.0 | 0.0 |
| Politicians | 0.1 | 0.1 | 0.1 | 0.1 | 0.6 |

### 3.1.1. Solidarities among Actors

Table 4 presents the solidarities among the different actors. The only actors offering solidarity with other actors are, on the one hand, Politicians, who express a positive 0.1 solidarity with each of the other actors, and on the other hand, Administrators, who offer 0.1 solidarity with Publishers.

*Solidarity of Politicians with all of the other actors (*Table 4*)*: Let us start with Politicians, a type of actor that should express solidarity with all its constituents. In the context of this model, this is represented by spreading around the value assigned by the actor for being supportive of the goals of actors other than themselves. In Scenario 0, we assigned the value 0.1 (10%) to the solidarity directed by this actor toward each of the other four, and 0.6 to itself, in order to represent politicians who want others to benefit, but not more than they themselves are, reflecting a very conservative stance. The reason for this distinctive behavior from politicians, however, is that they must express solidarity with all members of society, because their positions in charge depend on public opinion and electoral support.

*Solidarity of Administrators with Publishers*: Another interesting case of solidarity assignment occurs with that expressed by Administrators towards Publishers, where Administrators are the actor that represents the librarian bureaucracy inside research institutions. Some traditional librarians assume the defense of toll access to be the right way in which the system must be organized. This position seems to acknowledge that publishing takes effort, and therefore has some inherent costs that must be covered for the whole system to exist, none of which is denied by the OA position. To reflect this complex position, we assigned 10% to Publishers, of the total amount of solidarity to be assigned by Administrators. It must be noted that this reflects a particular stance based on what has been observed in local communities. However, there is evidence of the submissive behavior of Latin American Administrators with respect to global Publishers. As [39] explains: "In Latin America, several journals of good quality have been taken ahead by dedicated individual efforts and

under limited budgets, due to the lack of a subscription market and the weak support from governmental agencies, as opposed to the large amount of money dispensed to subscribe access to the journal collections commercialized by Publishers from developed countries".

### 3.1.2. Stakes and Effect Functions

Appendix B describes the technical issues regarding stakes and effect functions. In Table A1 of Appendix B, the actor in the *i* column controls the relational resource in the *i* row. This table provides the stakes committed by the actors to the resources, and the effect of the resources on the actors. Appendix B gives examples of the behavior of actors corresponding to the values of the state of the resources they control (see also Table 3, above), and provides a better explanation of stakes and effect functions.

The dependency of an actor on a resource is expressed by its stake and an effect function. An actor's stake in a resource expresses how much it needs this resource to achieve its goal. The effect function indicates how much the resource sways the actor. The total impact of a resource on an actor is the product of the stake and effect function. The addition of the impact of all resources on an actor defines the capability of the actor (see the previous section). For example, in the context of the model presented in this paper, from the Academics' perspective, the distribution of stakes would reflect the order of importance of resources for realizing their goal, which is to achieve prestige and acknowledgment in their research community, while for Publishers, it reflects the order of importance of the resources for increasing their monetary gains from publications and increasing the size of their publications. However, given the domain of possible values of a resource, the effect function indicates the effect of the resource on the actor for each value in its domain, which could be either positive or negative. For instance, high support from Politicians for open access (i.e., a high value of DemandOA) will have a strong negative effect on Publishers, but a strong positive effect on Academics. Conversely, low support from Politicians for open access will generate the opposite effects.

## 4. Results of the Simulations and Scenario Analysis

This section presents the results of the simulation and an analysis of the structures of all of the scenarios. In the first subsection, the results of the simulation of Scenario 0 are examined, and its structure is analyzed. In the following subsection, a scenario analysis (of the simulation and structure) of the other four scenarios is performed.

### *4.1. Scenario 0*

#### 4.1.1. Simulation of Scenario 0

Scenario 0 was simulated, and the mean values and standard deviations of the states of the resources and satisfaction of the actors were collected. Seventy replications were performed. The results of this scenario with the regulated configuration are shown in Tables 5–7, and then compared with the results of other configurations of Scenario 0 and with the other scenarios in the subsequent tables. All of the results are presented as the average values of the variables obtained for the replications of the experiments: Table 5 shows the values of the resources and the number of simulation steps; Table 6 presents the levels of satisfactions and influence; and Table 7 indicates the percentage values of maximum satisfaction and influence (to be explained below). To facilitate the analysis of the results, some of the results are repeated below: Table 8 (in column 10) presents (again) the value of the resources and the degree of satisfaction of the actors in this configuration of Scenario 0; Table 9 (column Sc-0) indicates (again) the values of the resources and the number of simulation steps of the regulated state; and Table 10 (columns Sc-0) presents (again) the values of satisfaction and influence for the regulated state of Scenario 0. The standard deviations are low, as indicated in Tables 5 and 6. The duration of the simulation ended up being long (more than 30,000 steps per replication, see the end of Table 5), which is coherent with the high amount of conflict present in the game between Publishers, on the one hand, and the rest of the player/actors, on the other hand. This conflict can be

noticed by observing the different effects of the resources on Publishers and the other actors (Table A1 of Appendix B), since the curves impact them differently; generally, what is good for Publishers is not good for the other actors, and vice versa.

**Table 5.** Averages of and deviations in the states of resources (regulated state).

| Resources | Averages | Deviations |
|---|---|---|
| PublishOA | 8.05 | 1.94 |
| SupportOA | 1.84 | 2.68 |
| DemandOAPublications | 6.08 | 2.38 |
| GrantOA | −2.13 | 1.12 |
| DemandOA | 9.97 | 0.06 |
| Num of simul. steps | 31,540.66 | 2298.34 |

**Table 6.** Averages of and deviations in level of satisfaction and influence of the actors (regulated state).

| Actors | Satisf. Averages | Satisf. Deviations | Influence Averages | Influence Deviations |
|---|---|---|---|---|
| Academics | 43.1 | 4.12 | 23.18 | 3.93 |
| Administrators | 30.4 | 3.1 | 3.55 | 1.13 |
| Funders | 79.74 | 3.92 | 58.17 | 3.28 |
| Publishers | −5.17 | 4.63 | 27.31 | 6.38 |
| Politicians | 65.76 | 2.69 | 101.37 | 0.91 |

**Table 7.** Percentage of total satisfaction and influence in the simulation. The table shows the percentage of total satisfaction received by an actor in the columns and total influence given in the rows, in relation to the total amount the actor can receive (give), both from (to) each other actor, and the general total is given in the penultimate right-most column (penultimate row). The total percentage of total auto-satisfaction (auto-influence) is shown in the right-most column (bottom row).

| | Acad. | Admin | Funders | Publish. | Polit. | Satisf. | Auto-Sat. |
|---|---|---|---|---|---|---|---|
| Academics | 99.1% | 0.0% | 80.5% | 39.5% | 100.0% | 75.4% | 19.5% |
| Administrators | 9.5% | 85.1% | 19.5% | 45.5% | 100.0% | 73.9% | 25.5% |
| Funders | 90.5% | 0.0% | 99.6% | 39.5% | 100.0% | 91.7% | 36.1% |
| Publishers | 9.5% | 41.0% | 19.5% | 95.7% | 0.0% | 48.5% | 78.0% |
| Politicians | 91.4% | 99.4% | 98.8% | 80.2% | 100.0% | 95.4% | 37.8% |
| Influence | 94.7% | 100.0% | 96.8% | 83.4% | 100.0% | | |
| Auto-Influence | 46.6% | 217.4% | 51.6% | 93.3% | 26.8% | | |

From Table 5, it can be seen that most of the resources either support open access or are in a moderate position, but none of them are strongly in favor of toll access: the highest levels of support for open access, in decreasing order of magnitude, come from Politicians, Academics and Funders, while Administrators are only slightly in favor of this, and Publishers show a minor preference for toll access.

**Table 8.** State of resources and satisfaction for several configurations of the model. The table shows the state of the resources (first five rows), the satisfaction of all actors (rows 5 to 10), and global satisfaction (row 11) for the following configuration of Scenario 0: state of maximum and minimum global satisfaction (at which the total sum of the satisfaction of the actors reach the maximum and the minimum values, respectively) (columns 1 and 2), and actor's maximum satisfaction (columns 3–7), regulated state (column 8), Nash equilibrium (column 9).

|  | Max-Glob | Min-Glob | Acad-Max | Admin-Max | Fun-Max | Pub-Max | Pol-Max | Simul | Nash-Equil. |
|---|---|---|---|---|---|---|---|---|---|
| PublishOA | 10.0 | −10.0 | 10.0 | −10.0 | 10.0 | −10.0 | 10.0 | 8.05 | 10 |
| SupportOA | 1.0 | −10.0 | −10.0 | 9.0 | −10.0 | −10.0 | 3.0 | 1.84 | 9 |
| DemandOAPub | 10.0 | −10.0 | 10.0 | −10.0 | 10.0 | −10.0 | 10.0 | 6.08 | 10 |
| GrantOA | 3.0 | −10.0 | 10.0 | 10.0 | 10.0 | −5.0 | 4.0 | −2.1 | −5 |
| DemandOA | 10.0 | −9.0 | 10.0 | 10.0 | 10.0 | −5.0 | 10.0 | 9.97 | 10 |
| Academics | 69.0 | −97.2 | 90.0 | 12.1 | 90.0 | −82.2 | 72.0 | 43.7 | 45 |
| Administrators | 43.4 | −77.8 | 31.8 | 72.6 | 31.8 | −60.9 | 48.6 | 31.6 | 27.1 |
| Funders | 89.5 | −97.2 | 96.5 | −8.2 | 96.5 | −92.2 | 90.5 | 80.3 | 81.5 |
| Publishers | −34.1 | 84.9 | −70.0 | −38.5 | −70.0 | 97.3 | −43.9 | −4.4 | −20.5 |
| Politicians | 73.3 | −76.6 | 69.4 | −4.3 | 69.4 | −48.7 | 73.5 | 66.5 | 64.5 |
| GLOBAL | 241.1 | −264.0 | 217.7 | 33.6 | 217.7 | −186.7 | 240.6 | 217.7 | 197.6 |

**Table 9.** Mean values of the states of the resources (first five rows) and the number of simulation steps required for the configuration of the regulated state (last row) for all scenarios (Sc-i, where i = 0 to 4).

|  | Sc-0 | Sc-1 | Sc-2 | Sc-3 | Sc-4 |
|---|---|---|---|---|---|
| PublishOA | 8.05 | 3.15 | 8 | 9.08 | 5.83 |
| SupportOA | 1.84 | 2.83 | 10 | 9.5 | 1.64 |
| DemandOAPub | 6.08 | −0.87 | 10 | 9.9 | 8.09 |
| GrantOA | −2.13 | −2.19 | −10 | −9.97 | −1.14 |
| DemandOA | 9.97 | 9.95 | 10 | 10 | 9.97 |
| Num of simul. steps | 31,540.66 | 24,038.5 | 39,102.36 | 38,966.89 | 14,375.8 |

**Table 10.** Mean values of satisfaction (5 left-most columns) and influence (5 right-most columns) for all actors and globally (rows) for the regulated configuration, and for all scenarios (Sc-i, where i = 0 to 4).

|  | Satisfaction | | | | | Influence | | | | |
|---|---|---|---|---|---|---|---|---|---|---|
|  | Sc-0 | Sc-1 | Sc-2 | Sc-3 | Sc-4 | Sc-0 | Sc-1 | Sc-2 | Sc-3 | Sc-4 |
| Academics | 43.1 | 22.9 | 29.2 | 29.7 | 50.7 | 23.18 | 1.7 | 23.7 | 25.5 | 21.3 |
| Administrators | 30.4 | 34.4 | 11.8 | 11.7 | 34.2 | 3.55 | 4.5 | −2.6 | −2.1 | 4.7 |
| Funders | 79.74 | 0 | 72.7 | 74.7 | 66.4 | 58.17 | −0.9 | 62.8 | 62.9 | 64.7 |
| Publishers | −5.17 | 12.2 | −16.6 | −17.8 | 14.0 | 27.32 | 26.9 | −37 | −37 | 31.8 |
| Politicians | 65.76 | 7.2 | 51.5 | 52.8 | 65 | 101.37 | 44.5 | 101.9 | 101.9 | 107.9 |
| GLOBAL | 213.83 | 76.7 | 148.6 | 151.1 | 230.3 |  |  |  |  |  |

The most satisfied actors (see Table 6) are, in order of intensity of satisfaction, Funders, Politicians, and Academics, followed by Administrators, and finally Publishers. This is an indicator of the capacity actors receive from others. Publishers have a very limited capability to realize their own goals (depending on themselves). The opposite is true

for Politicians and Funders, while Academics and Administrators are in an intermediate position. Similarly, the most influential actors are Politicians and Funders, followed by Publishers and Academics. Administrators have low influence. This is partially consistent with the relevance of the resources controlled by the various actors, which are as follows in decreasing order: Politicians, Publishers, Funders, Academics, and Administrators, as indicated above and observed on the right-hand side of Table A1 in Appendix B. The case of Publishers does not correspond with this order: Publishers have too little influence, which is not in agreement with the high relevance of the resource they control. This can be explained by the fact that Publishers are in conflict with the rest of the actors and do not collaborate completely with them, as Publishers pursue their own interests. Publishers reach (regarding the state of the resources) some level of satisfaction by minimumly expressing their preference for toll access, as this resource is slightly below zero (zero represents the middle ground between granting OA and offering toll access).

Thus, when measured on the basis of satisfaction, the actors with the best position in the game are Politicians and Funders, while those in the worst position are Publishers (low satisfaction and low capacity to reach their goals), and Administrators (which also have too low an influence). Academics are in an intermediate position. In particular, Publishers seem to receive strong pressure from the other actors. This is reflected in Table 7, which shows that Publishers obtain only 48.5% of the total satisfaction they could with the current structure of the model, while giving themselves 78% of the total satisfaction they are able to obtain (auto-satisfaction) (see the right side of Table 7).

Table 7 also indicates that Publishers obtain 83.4% of the total influence they are able to exert, while giving themselves 93.3% of their possible total auto-influence. This means that Publishers cannot achieve high levels of influence within the actual structure of the game. This offers a general idea of the degree of conflict they maintain with other actors, along with a measure of their isolation.

### 4.1.2. Analysis of the Structure of Scenario 0

Any assignment of resource values gives a certain configuration of the game. Above, we studied the results for a single configuration of the game, represented by Scenario 0, which is the configuration given by the regulated state (simulation). The analysis of the structure of each scenario allows us to better investigate, on a globally level, some of the properties of the sub-model representing each scenario. In this subsection, the structure of Scenario 0 is analyzed to better understand the constraints and characteristics of the game by analyzing some of the configurations of the game represented by this scenario. This complements the previous results and provides a wider view of the structure of the game represented by Scenario 0. Although the game has infinite possible configurations, only some of them are of particular interest for further understanding the game; for instance, the configuration with maximum total satisfaction (i.e., the sum of the satisfaction of all actors), the configuration in which the satisfaction of a certain actor reaches its maximum value, and the configuration with Nash equilibrium. Table 8 displays the values of the resources and global satisfaction of the actors (rows) for the following configurations (columns): max. global satisfaction, min. global satisfaction, max. satisfaction of each actor, the regulated state (simulation), and Nash equilibrium. It also provides the total number of simulation steps (last column) for these configurations of Scenario 0.

Table 8 shows that, at the configuration of maximum satisfaction for Publishers (right-most column), the other actors have low levels of satisfaction, especially Funders and Academics, followed by Administrators and Politicians. Correspondingly, as expected, for the configurations in which each of these actors has the maximum level of satisfaction, Publishers have low levels of satisfaction, followed by Administrators and Politicians, especially for the configurations in which Funders and Academics have the maximum levels of satisfaction.

This suggests a circumstantial alliance between Funders, Academics, and Politicians for facing common conflict with Publishers. This circumstantial common interest is espe-

cially relevant for the following pairs of actors: Funders–Academics, Funders–Politicians, and Academics–Politicians. Administrators occupy a somewhat intermediate position, although a bit closer to the group with those three (Funders, Academics and Politicians). This is coherent with the state of the resource controlled by Administrators (the resource receives a value of 1), which is above zero, and therefore indicates them to be somewhat in favor of open access.

In addition, Politicians are the actors whose situation best resembles the global best situation of the game. Among the configurations in which the satisfaction of each actor is maximized, the configuration in which the satisfaction of Politicians is maximized offers the highest value of global satisfaction (240.6). Furthermore, this value is very close to the global maximum satisfaction value (241.1). These conclusions are confirmed by (and are consistent with) the results corresponding to the configurations in which the actors' satisfaction is minimized (not shown here).

Finally, Table 8 also indicates that the configuration of the regulated state of Scenario 0 is somewhat between the configuration of maximum individualism, or Nash equilibrium, and the state of maximum global satisfaction (217.7; the global satisfaction for the configuration of the regulated state is between 197.6 and 241.1, which are the values of global satisfaction at the configuration of the Nash equilibrium and maximum global satisfaction, respectively). This will be confirmed in Section 4.2.2 by the fact that the distance between the configuration of the regulated state and the other two configurations is very similar.

### 4.2. Comparison of Scenarios

As alternatives to Scenario 0, the alternative scenarios defined above (Scenarios 1–4) are studied in this section to evaluate the possible representations of realistic and hypothetical cases of interest regarding the promotion of open access.

### 4.2.1. Simulation of Scenarios 1–4

For all scenarios and the configuration of the regulated state, the values of the resources and the length of the simulations are presented in Table 9, whereas the values of satisfaction and influence are summarized in Table 10.

The most important differences in the resources of these scenarios with respect to those in Scenario 0 are as follows:

- Scenario 1: Resources controlled by Academics and Funders are decreased. This generates an important reduction in the satisfaction of all actors, except Administrators, whose satisfaction only varies slightly, and Publishers, for whom it increases. The capacity of Publishers to reach their own goals increases (this is the most remarkable result), while, as expected, that of Politicians and Funders decreases. Similar results are obtained for influence, as the influence of most of the actors decreases, but that of Academics only varies slightly, and that of Publishers does not vary. The number of simulation steps falls by one-third with respect to the number in Scenario 0, which indicates a reduced amount of conflict in the game. Consequently, Publishers have a better position in the game, while Academics, Politicians and Funders have a worse position than in Scenario 0, which is linked to the disengagement of Funders and Politicians, who are less active in this game.

- Scenarios 2 and 3: In these scenarios, the extremist position of Publishers in favor of toll access increases the number of simulation steps compared to Scenario 0, suggesting that the game presents a stronger degree of conflict than that scenario. All other actors also take an extremist position, now strongly supportive of open access. Academics, Administrators and Funders join the position of Politicians in favor of open access. The satisfaction of all actors, except Funders, decreases (for this scenario, the regulated configuration is very close to the Nash equilibrium configuration, as we will see below). Remarkably, the capacity of Publishers to reach their goals decreases. The influence of three actors—Academics, Funders and Politicians—does not vary significantly compared to Scenario 0, while that of Administrators and Publishers

decreases, especially that of Publishers. Thus, Publishers are in a worse position when taking an extremist position against open access than when they are more permissive, as occurs in Scenario 0.

- Scenario 4: Looking at the results in Tables 9 and 10, the solidarity of Funders with Academics seems to reduce the amount of conflict, as suggested by the significant decrease in the number of simulation steps. Interestingly, the regulated configuration of Scenario 4 was close to that of Scenario 0. Surprisingly, while Funders increase their support for open access (the resource they control rises), Academics decrease theirs, and Publishers slightly augment theirs. Academics are now satisfied with less effort than in Scenario 0, and consequently, Publishers relax slightly. It seems that it is difficult for all actors to reach a better state of satisfaction than in Scenario 0, as the game is very stable; in Scenario 4, while Funders work partly in favor of Academics, Academics somewhat disengage from the game, which in turn causes Publishers to relax, so that in the end, the amount of conflict decreases. Funders' level of satisfaction undergoes a small reduction, as expected, since they pursue their own benefits only only to an extent of 60%, while that of Academics slightly increases, and, surprisingly, that of Publishers undergoes a more significant increase than that of Academics. Meanwhile, the level of satisfaction of Publishers and Administrators does not vary. Compared with Scenario 0, similar to what occurred in Scenario 1, the capacity of Publishers to realize their goals increases. This means that the increase in the solidarity of Funders with Academics unexpectedly has a relevant impact on Publishers, making them a bit more permissive regarding open access, increasing their level of satisfaction and capacity (the value of their resource increases to a small degree, augmenting their support for open access). Influence undergoes only small changes, especially in the following cases: the influence of Funders and Publishers undergoes a small increase, while the influence of Academics slightly decreases.

### 4.2.2. Analysis of the Results Given the Structures of Scenarios 1–4

Table 11 presents the state of the resources (the first five rows), the satisfaction of the actors (the next five rows), and the maximum global satisfaction (the last row) for the following configurations of each scenario (in the columns): maximum global satisfaction (MG), minimum global satisfaction (mG), Nash equilibrium (Ne), and regulated state (Re).

In Scenarios 1–4, similar to Scenario 0, when Publishers assume an extremist position in favor of toll access (against open access), an alliance between Academics, Funders and Politicians against Publishers is observed, especially in Scenarios 2 and 3. In the configuration of the regulated state (see Table 11), Administrators are neutral in those scenarios in which Publishers do not assume an extremist position in favor of toll access, in the sense that they slightly support OA, and their position (management of their resource) is consistent with that shown in the configuration in which maximum global satisfaction is achieved (despite the conflictive situation, they somewhat contribute to common benefit). Conversely, when Publishers assume an extreme position in favor of toll access, Administrators strongly support OA by setting the resource they control (SupportOA) close to the value corresponding to the Nash equilibrium configuration. This circumstantial separation between the positions of Administrators and Publishers in Scenarios 2 and 3 is clearly negative for Publishers, as their satisfaction strongly decreases. This is related to the more egoist stance of the actors, as the regulated state is closer to the Nash equilibrium in which each actor only takes care of their own satisfaction.

**Table 11.** Max. global (MG), min. global (mG), Nash (Ne) and regulated (simulation) (Re) equilibrium configurations for Scenarios 1–4 (columns). The state of the resources, the satisfactions of the actors, and global satisfaction are displayed in rows, for the configurations Re, MG, gG and Ne of Scenarios 1–4 (columns).

| | | Scenario 1: Politicians and Funders Are "Inactive" | | | | Scenario 2: Publishers Are Less Permissive Regarding Open Access | | | | Scenario 3: Publishers Are Extremely Less Permissive Regarding Open Access | | | | Funders Direct 40% of Their (Positive) Solidarity towards Academics | | | |
|---|---|---|---|---|---|---|---|---|---|---|---|---|---|---|---|---|---|
| | | **MG** | **mG** | **Ne** | **Re** | **MG** | **mG** | **Ne** | **Re** | **MG** | **mG** | **Ne** | **Re** | **MG** | **mG** | **Ne** | **Re** |
| States | PublishOA | −2 | 10 | 10 | 3.15 | 10 | −10 | 10 | 8 | 10 | −10 | 10 | 9.08 | 10 | −10 | 10 | 5.83 |
| | SupportOA | 1 | −10 | 9 | 2.83 | 1 | −10 | 9 | 10 | 1 | −10 | 9 | 9.5 | 1 | −10 | 9 | 1.64 |
| | DemandOAPub | 10 | −10 | −10 | −0.87 | 10 | −10 | 10 | 10 | 10 | −10 | 10 | 9.9 | 10 | −10 | 10 | 8.09 |
| | GrantOA | 2 | −10 | −5 | −2.19 | 10 | −10 | −10 | −10 | 10 | −2 | −10 | −9.97 | 10 | −10 | −4 | −1.14 |
| | DemandOA | 10 | −8 | 10 | 9.95 | 10 | −9 | 10 | 10 | 10 | −9 | 10 | 10 | 10 | −9 | 10 | 9.96 |
| Satisf | Academics | 55.9 | −69.8 | −5 | 22.9 | 90 | −97.2 | 30 | 29.2 | 90 | −73.2 | 30 | 29.7 | 90 | −97.2 | 48 | 50.7 |
| | Administrators | 42.7 | −81.4 | 30.1 | 34.4 | 60.2 | −76.6 | 11.6 | 11.8 | 60.2 | −59.4 | 11.6 | 11.7 | 60.2 | −77.8 | 30.2 | 34.2 |
| | Funders | 0 | 0 | 0 | 0 | 96.5 | −97.2 | 76.5 | 72.7 | 96.5 | −89.2 | 76.5 | 74.7 | 93.9 | −97.2 | 68.7 | 66.4 |
| | Publishers | −10.1 | 55.9 | 9.5 | 12.2 | −86.5 | 96.9 | −18.5 | −16.6 | −86.5 | 16.9 | −18.5 | −17.8 | −50.5 | 46 | −1.7 | 14 |
| | Politicians | 9.4 | −11.1 | 3.7 | 7.2 | 71.1 | −75.4 | 54 | 51.5 | 71.1 | −67.6 | 54 | 52.8 | 71.1 | −76.6 | 66.3 | 65 |
| GLOBAL | | 97.9 | −106.4 | 38.3 | 76.7 | 231.3 | −249.6 | 153.6 | 148.6 | 231.3 | −272.7 | 153.6 | 151.1 | 174.7 | −205.6 | 16.5 | 179.6 |

All of the following results, presented in Table 11, are confirmed when analyzing the configurations in which minimum global satisfaction and minimum satisfaction per actor are achieved (results not shown in the paper): (i) the circumstantial alliance between Academics, Funders, and Politicians against Publishers, and the relatively neutral state of Administrators, in Scenarios 0 and 4; (ii) the alliance between all of them in Scenarios 2 and 3; and (iii) the similarity between Scenarios 0 and 4.

To provide a more precise account of the configurations, Table 12 compares, for each scenario, the configuration of the regulated state with the configurations in which the maximum global satisfaction and Nash equilibrium are obtained, considering the following: (a) the distance between the configurations in terms of the state of the resources (DState); and (b) the difference between the configurations with respect to global satisfaction (DSat).

**Table 12.** Distance between the simulation or regulated configuration (Re) and the configurations in which global maxima and Nash equilibrium are achieved for all scenarios. For each scenario (columns) we have: (i) (right side) the distance (DState) (measured as the Euclidean distance between the points given by the state of the resources for each configuration) from the regulated configuration (simulation) to (a) (upper row) the configuration in which the global maximum satisfaction is achieved and (b) (lower row) the configuration in which the Nash equilibrium is achieved; and, (ii) (left side) the difference (DSat) between the total level of satisfaction achieved with the regulated state configuration and (a) the total level of satisfaction achieved with the configurations in which maximum global satisfaction is achieved (upper row) and (b) the configurations in which the Nash equilibrium is achieved (bottom row).

| | Scenario 0 | | Scenario 1 | | Scenario 2 | | Scenario 3 | | Scenario 4 | |
|---|---|---|---|---|---|---|---|---|---|---|
| | DState | DSat | DState | DSat | DState | DSat | DState | DSat | DState | DSat |
| Glob-Max | 6.78 | −23.4 | 12.87 | −21.2 | 21.91 | −82.7 | 21.72 | −80.2 | 12.06 | 4.9 |
| Nash Equi | 8.88 | 20.1 | 13.28 | 38.4 | 2.03 | −5 | 1.05 | −2.5 | 9.13 | 16.1 |

It can be observed that for all scenarios, except Scenarios 2 and 3 (where Publishers have an extreme position favoring toll access, i.e., against open access), the configuration of the regulated state is intermediate between the configurations of maximum global satisfaction and the Nash equilibrium. For Scenarios 2 and 3, the configuration of the regulated state is much closer to the configuration in which the Nash equilibrium is obtained than to the configuration in which maximum global satisfaction is achieved. In accordance with this, the more extremist the position of Publishers (Scenario 3), the closer the regulated state configuration is to the Nash equilibrium configuration, in which Publishers and several other actors are in a worse position than in Scenarios 0 and 1. Thus, it is in the interest of Publishers not to take an extremist position against open access. Comparing scenarios 0, 1, and 4, the regulated configuration of Scenario 4 is the farthest from the Nash equilibrium, followed by scenarios 0 and 1. Ordering the scenarios by actors' individualism from lower to higher, we have Scenarios 4, 0, 1, 2, and 3.

All this allows us to better comprehend the game of open access: the most favorable configurations for open access are those belonging to Scenario 4 and Scenario 0. The regulated state of Scenario 1 is farther from the Nash equilibrium than those of Scenarios 0 and 4, but its regulated state is distant from that of global maximum satisfaction, while Academics' satisfaction and influence are lower than in Scenarios 0 and 4. In Scenarios 0 and 4, Academics have the best satisfaction; power and satisfaction are somewhat better distributed, and conflict is usually decreased compared to the other scenarios. Scenario 4 achieves the greatest maximum global satisfaction in the regulated state. Following this criterion, the complete order of goodness of the scenarios for open access is Scenario 4, Scenario 0, Scenario 1, and Scenarios 3 and 2.

## 5. Discussion

The results of Scenario 0 show that, in the model, Publishers are subjected to strong pressure from Academics, Politicians and Funders, being isolated in their interests in the driving conflict of the game. The Administrators seem to adopt an intermediate stance. To address this conflict, Publishers do not assume an extreme attitude against open access as their best strategy. When Administrators and Publishers take on those "intermediate" positions, this situates the configuration of the regulated state between the configurations of maximum global benefit and the Nash equilibrium.

In Scenario 1, representing the game in regions and situations of Latin America, Africa, most of Asia, and many other areas of the world, where Politicians and Funders are disengaged from the game of open access, Academics are more isolated than in Scenario 0, finding certain circumstantial alliances only with Administrators. Administrators still assume an intermediate position between supporting open and toll access, while Publishers are in a much better position than in Scenario 0 (pressure over Publishers decreases as the conflict reduces in their favor). Administrators and Publishers continue to adopt intermediate positions. A possible lesson is that to move all of the games in favor of OA, as in Scenario 0, Politicians and Funders must be motivated to increase their commitment in favor of open access, thus becoming more active. This could be achieved, for example, by improving their awareness of their own benefits from open access, such as the savings obtained on pay-to-publish budgets.

It is not in the interest of Publishers to assume an extreme position against open access (Scenarios 2 and 3), as this will break the neutral position of Administrators, who would support open access, and will decrease their satisfaction and action capacity, since all of the other actors would constitute a circumstantial block in favor of open access, contrary to the interests of Publishers. Clearly, in these scenarios, the actors become more individualist, as the regulated state is closer to the Nash equilibrium than in the other scenarios.

When the solidarity of Funders with Academics increases to 40% of the total that Funders can give (Scenario 4), Funders' support for open access increases, but, surprisingly, Academics reduce their backing for open access. This situation reduces the conflict in the game compared with Scenario 0, favoring Publishers, who become slightly more permissive about open access, undergoing a certain increase in their satisfaction and action capability. This scenario and Scenario 0 are the best situations for open access, as most of the actors are well satisfied, have good action capacity, and with respect to the other scenarios, power and satisfaction are better distributed and the amount of conflict decreases. Scenario 4 is also the best scenario for Publishers, which is a counterintuitive result considering that Academics, Funders and Politicians maintain their alliance for open access.

In accordance with this, and given that the model is a representation of a realistic situation, as interpreted from working experience and activism of domain experts, the study offers some suggestions for actions. Among these suggestions, we find that in order to promote open access in general, a higher involvement in the game in favor of open access by Funders and Politicians seems to be required. Additionally, a more flexible position of Publishers with respect to open access could also be useful for the sake of efficiency in the game (e.g., by considering and observing the benefits of some alternative strategies and actions, such as hybrid publications, Publishers allow other actors to benefit more).

Up until this point, we have not considered certain important public policies for funding in the US and Europe that will transform the OA game, especially the OSTP "Nelson memo" of August 2022 in the US [40][1] and Plan S in Europe [41]. This will lead to a new equilibrium in the academic publishing game. We already observe strong support for open access in Europe and North America, as noted above, but these policies will provoke a new configuration of the structure of the game in favor of the total prevalence of OA. In a new configuration of the game, Funders would have to permit grantees to pay potentially higher Article Processing Charges (APCs), but these increases in grant budgets would, in time, be offset by decreased expenses to university libraries for journal subscriptions. Major

Publishers seeking profit would have to find income streams other than subscriptions to maintain their financial stability.

Despite this, we believe that this paper will help readers to better understand the actual OA game and compare OA practices against diverse cultural backgrounds. This study is based on intuitions and expert domain knowledge, which are used to interpret some historical and situational contexts for the game. This represents an alternative analytical strategy, and is even a complement for a historical and situational review of the real game.

## 6. Conclusions

In this study, we examined the dynamics of the cause of open access activism, which aims to provide unrestricted electronic access to published scientific products. By reducing the complexity of the system to a game of interest and focusing on the behaviors and motivations of the collective actors involved, we gained insights that allowed us to make suggestions for the actors in the game based on expert domain knowledge obtained through activism.

The model confirmed certain intuitions and shed light on the relationships between the actors. Publishers were found to be somewhat isolated in their interests compared to other actors. The model demonstrated the potential for circumstantial alliances against Publishers and showed that strong opposition to open access is not in the best interest of Publishers.

We developed several scenarios, representing both realistic and hypothetical situations, to examine how open access can be promoted. The order of goodness for open access was found to be Scenario 4, Scenario 0, Scenario 1, Scenario 3, and Scenario 2. Goodness of open access is defined as a high degree of satisfaction among Academics, a decreased amount of conflict in the game, and a good distribution of power and satisfaction among the actors. The decrease in the amount of conflict and the degree of individualism of actors was measured by observing the distance from the regulated state to the Nash equilibrium for each game configuration.

The lessons learned from experiments investigating how each scenario promotes or hinders open access can be summarized as follows:

- Scenario 0. This scenario represents the base model that can be observed in most of Europe and North America. It exhibits a medium level of conflict, a moderate degree of satisfaction for Academics, a high degree of satisfaction for Politicians and Funders, and a low degree of satisfaction for Publishers.
- Scenario 1. This scenario cancels out Politicians' and Funders' interest in the resources, indicating their high level of disengagement from the game. Compared to Scenario 0, the quality of open access decreases, including a reduced level of satisfaction and influence on the part of Academics.
- Scenarios 2 and 3. These scenarios represent a more negative perception of Publishers toward open access compared to Scenario 0. In these scenarios, the negative effect of open access on Publishers is stronger, particularly in Scenario 3. Publishers do not consider alternative actions or strategies to either mitigate the negative effects of or gain benefits from open access, such as engaging in hybrid publications. In these cases, the goodness of open access decreases, Academics' satisfaction decreases, the amount of conflict in the game increases, and the regulated state approaches the Nash equilibrium.
- Scenario 4. In this scenario, Funders' solidarity with Academics is increased compared to the base model. As a result, open access is further promoted when compared to Scenario 0. Academics experience increased levels of satisfaction and reduced amounts of conflict. Paradoxically, Academics participate less in open access (their resource, open publications, is lower), but they receive better benefits in terms of level of satisfaction than in Scenario 0.

In summary, open access benefits from high degrees of involvement of Funders and Politicians in the game (as is the case in Scenario 0), increased solidarity of Funders with

respect to Academics (as is the case in Scenario 4), and a flexible disposition on the part of Publishers to exploring alternative strategies and actions in order to benefit from open access (as is the case in Scenario 0, in contrast to Scenarios 2 and 3).

Scenario 0 represents the manner in which the open access game operates in Europe and the USA, while Scenario 1 represents that in regions like Latin America and many countries in Asia and Africa, where Politicians and Funders are more distant from the game. The results confirm that open access is in a better situation in Europe and North America compared to Latin America, Asia, and Africa. However, there have been interesting but isolated advances in Latin America, such as Redalyc and SciELO [42]. Increasing awareness among Funders and Politicians about the societal benefits of open access in these regions (represented by Scenario 1) could promote open access and bring it closer to the situation described in Scenario 0. This would be complemented by a more permissive and flexible stance of Publishers with respect to open access.

**Author Contributions:** Conceptualization, J.D. and O.T.; Methodology, O.T. and J.D.; Software, O.T. and J.D.; Validation, J.D. and O.T.; Formal Analysis, O.T. and J.D.; Investigation, J.D. and O.T.; Resources, O.T. and J.D.; Data curation, O.T. and J.D.; writing—original draft preparation, O.T. and J.D.; Writing—review and editing, J.D. and O.T.; Visualization, O.T. and J.D.; Supervision, O.T. and J.D.; Project administration, J.D. and O.T.; All authors have read and agreed to the published version of the manuscript.

**Funding:** This research received no external funding.

**Data Availability Statement:** Readers interested in replicating results or testing alternative scenarios will find the sources (the module of the model) available at: https://github.com/oteran22/SimulatingTheGameOfOpenAccess-SocLab. SocLab is freely available from https://sourceforge.net/projects/soclab/).

**Conflicts of Interest:** The authors declare no conflict of interest.

## Appendix A. Bases of Social Simulation with SocLab

The SocLab framework [43,44] was developed as a formalization of the Sociology of Organized Action (SOC) of Crozier and Friedberg [27]. SOC examines how social organizations are regulated through power relationships between actors. It recognizes that actors possess bounded rationality and are influenced by their complex environment, shaping their behavior [45].

In SOC, power relationships are based on actors' control of resources that are needed by others. Actors follow strategies in social games defined by their interactions with others. They seek to achieve a satisfying level of capability to reach their goals, balancing the aims assigned to their role within the organization or social interaction and their individual objectives. The SocLab simulation platform [24,46] allows for the definition of an organization's structure, the analysis of its structural properties, the exploration of possible configurations (Pareto optima, Nash equilibrium, structural conflicts and others), and the computation of actor behavior within the organizational context.

The simulated social game is deemed to be solved when it reaches a stationary/regulated state, i.e., a configuration where no actor has reasons to change his behavior, i.e., where for all actors the satisfaction of the actor is greater than their aspiration.

The management of a resource determines how it can be used by others and forms the basis of the relationship between the manager and the users of the resource. An actor's dependency on a resource is expressed through its stake and effect functions (as shown in Table A1 of Appendix B, for Scenario 0). The stake represents the actor's dependence on the resource (on a scale null = 0, negligible = 1, . . ., significant = 5, . . ., critical = 10), while the effect function describes the extent to which the management of the resource by the controlling actor impedes or supports the achievement of goals by the recipient actor.

As a way to address the complexity, each actor is allowed to allocate only 10 points on stakes to distribute over the set of all resources.

The impact of a resource "r" with management "m" on an actor "a" is determined by the effect$_{r, a}$() function applied to "m," weighted by the stake of "a" on "r" (stake(a, r)). The satisfaction of an actor is an aggregation of the impacts it receives from the resources it depends on, weighted by the solidarities it extends to other actors. Solidarities measure the degree of concern or support an actor has for another.

The x-axis of the effect$_{r, a}$() functions corresponds to the state of the resource, i.e., how it is managed, from the least (negative values) to the most (positive values) cooperative way. The y-axis corresponds to the actor's capability to achieve its goals, depending on the way the resource is being managed.

The satisfaction of actor "a" (satisfaction(a, m)) is defined as the sum of the solidarities(a, b) (solidarity between actors "a" and "b"), multiplied by the capability(b, m) (capability of actor "b" achieved through the resources it depends on). The influence exerted by actor "a" on actor "b" (influence(a, b, m)) is the sum of the solidarities(b, c) (solidarity between actors "b" and "c") multiplied by the capability(a, c, m) (capability of actor "a" provided to actor "c" through resource management).

The influence of actor "a" (influence(a, m)) is the sum of the influences it has on all other actors. The SocLab decision-making algorithm seeks to maximize actor satisfaction. Each actor maintains an aspiration variable that approximates its actual satisfaction. When an actor's satisfaction exceeds its aspiration, a stationary state is reached, indicating a regularized state where actors perceive their satisfaction as the best they could achieve in accomplishing their goals.

Formally, it can be said that, when a system of organized action is in a configuration $m = (m_r)_{r \in R}$, the capability obtained by an actor $a$ is defined as:

$$\text{capability}(a, m) = \sum\nolimits_{r \in R} \text{stake}(a, r) \times \text{effect}_{r,a}(m_r) \tag{A1}$$

From this we can define the satisfaction of $a$ as (calling $A$ the set of all actors):

$$\text{satisfaction}(a, m) = \sum\nolimits_{b \in A} \text{solidarity}(a, b) \times \text{capability}(b, m) \tag{A2}$$

Conversely the influence exerted by an actor a over an actor b is defined as the sum of the capabilities it supplies to the actors towards which b has solidarity by the management of the resource(s) it controls, that is:

$$\text{influence}(a, b, m) = \sum\nolimits_{c \in A} \text{solidarity}(b, c) \times \text{capability}(a, c, m) \tag{A3}$$

Finally, the influence of $a$ is the sum of the influence of $a$ on all actors:

$$\text{influence}(a, m) = \sum\nolimits_{b \in A} \text{influence}(a, b, m) \tag{A4}$$

In the context of the open access game, a high satisfaction of Academics and a decrease in conflict, along with a well-distributed power and satisfaction across actors, define a favorable situation for open access.

## Appendix B. Description of the Scenarios

1. Scenario 0.

*Distribution of stakes:* Each actor allocates a normalized sum of 10 stake marks. The stake of the actor controlling the relationship is shown in bold in Table A1. The relevance column indicates the sum of stakes placed on each resource. The resources controlled by Politicians and Publishers are the most relevant, followed by those controlled by Funders (intermediate relevance). Finally, we have the resources controlled by Academics and Administrators (especially those controlled by Administrators). This order is related to the potential to achieve influence. In accordance with this element, the potential of the actors for influence is in the following order: Politicians, Publishers, Funders, Academics, Administrators.

**Table A1.** Table of stakes, effect functions and relevance of each resource for Scenario 0. Each cell presents the interaction among actors mediated by the stakes and the effect functions: each cell shows (as a number) the stake of the actor (column) on the resource (row), and (as function) the effect of the resource on the actor's activity. The sum of the stakes by column is 10 (each actor distributes 10 stakes among the resources). Additionally, the last column indicates the relevance of each resource: the sum of the stake for the resource in each row.

| Effect | Academics | Administrators | Funders | Publishers | Politicians | Relevance |
|---|---|---|---|---|---|---|
| PublishOA | **2.0** | 0.0 | 2.0 | 1.5 | 2.0 | 7.5 |
| SupportOA | 0.0 | **3.0** | 0.0 | 1.5 | 0.0 | 4.5 |
| DemandOAPublications | 2.5 | 0.0 | **3.5** | 1.5 | 3.5 | 11 |
| GrantOA | 3.0 | 3.5 | 1.0 | **4.0** | 1.0 | 12.5 |
| DemandOA | 2.5 | 3.5 | 3.5 | 1.5 | **3.5** | 14.5 |

*Effect functions:* For each effect function (shown in each cell), the x-axis represents cooperativity when a resource is managed by its controlling actor, whereas the y-axis is the range of the resulting impediment or facilitation for the dependent actor. Low values of effect functions (horizontal edge) indicate that the resource favors toll access, while large values imply that the resource promotes open access: the lowest the value of a resource, the more it supports or corresponds to toll access. The shape of the effect function shows a conflict between Publishers and other actors; for Publishers, generally, the lower the value of the resources (i.e., if the resource favors toll access), the better the effect Publishers have, while the opposite is the common case for other actors.

Next, the distribution of stakes on the relationships of each actor and the effect functions are justified.

*Distribution of stakes of Academics:* Stakes model a form of scaling for the effects of the resources from some actor on the capability of another actor. This means that the effect of a resource on an actor can be magnified or reduced, depending upon the stake that the actor sets for that resource (controlled by the actor to which the stake is set). Academics are particularly sensitive to the behavior of Publishers, who control the means of publication. For this reason, we decided that Academics would assign the largest portion of the 10 stakes to the resource controlled by Publishers (3.0), as shown in Table A1. However, Academics, especially in public institutions, remain formally subordinated to national legislation, and Politicians should therefore be considered as representatives of people. Additionally, funding has become a strong guiding force. For these reasons, stakes of 2.5 were assigned to the resources controlled by Politicians and Funders. Academics' last scaling influence in this sense comes from themselves, as experiences are shared among colleagues. For this reason, we left 2.0 for Academics to assign to their own resources in Scenario 0.

*Distribution of stakes of Administrators:* Administrators represent the institutional bureaucracy, and therefore, the model in Scenario 0 assigns greater importance to the resources controlled by Politicians (3.5) and the Administrators themselves (3.0). However, to reflect the closest alliances between Librarians and Publishers, which sometimes see themselves

as colleagues, we set the corresponding value to 3.5. It is worth noting that Administrators in this scenario have no regard for Academics, who they consider less qualified to make decisions on where to publish, or for Funders, with whom Administrators normally have no direct interaction.

*Distribution of stakes of Funders:* Funders play the role of setting the agenda for research while paying for it. To reflect this leading intention, we set the highest assignment of their stakes to the resource they control, at 3.5. Funders, however, acknowledge national legislation in these nations in Scenario 0. Therefore, a similar stake is assigned to the resource controlled by Politicians. The remaining, lesser part of the stakes is distributed among those resources controlled by the actors who "do the job": the resource controlled by Academics received 2.0, and that managed by Publishers 1.0. This somehow reflects the relative importance of the role of these actors in the production of science, according to that worldview.

*Distribution of stakes of Publishers:* For Publishers, the most important effect is caused by the resources they control. To reflect this, we decided to set the stake at 4.0, leaving all the other actors' resources receiving the same level of stake (1.5 for each). The balance, however (6.0 for all other resources and 4.0 for its resource), captures the fact that Publishers are normally commercial enterprises that try to conform to consumers' needs.

*Distribution of stakes of Politicians*: Finally, Politicians have the same intention to lead as that described above for Funders. They also want to establish an agenda for research. Therefore, for Politicians, we decided to mirror the assignments of the Funders to all other actors.

*Effect functions of Publish Open Access (PublishOA) on the actors:* The function for this resource must reflect the fact that Academics, Funders and Politicians are happier with research being openly available after publication. The first because they will be able to access it themselves and share it indiscriminately. The latter because they will not have to pay to access it after it has been published. This, of course, goes directly against the traditional source of income for Publishers, which is why their effect function is decreasing monotonically. We also wanted to reflect on the curious fact that Administrators remain uneventful with either result.

*Effect functions of Support Open Access (SupportOA) on the actors*: We decided to reflect the total indifference of the Academics towards the effect of the resource controlled by Administrators, which is consistent with the stakes Academics assign to the resource (and which are also the same mutatis mutandis). The same factual indifference towards the resource occurs from Funders and Politicians. Administrators are assumed to be favored by increases in their own resources up to a point. Any support for OA from Administrators will hamper the benefits of Publishers.

*Effect functions of Demand Open-access publications (DemandOAPublications) on the actors*: For Scenario 0, the model assumes that the demand for OA Publications from the Funders of the research project is beneficial for academic institutions (Academics and Administrators), the public, and therefore for Politicians, as well as the Funders themselves, due to the savings for not having to pay again to read those publications. Only Publishers will be directly and correspondingly affected in a negative way by these demands. However, to reflect that there may be interests connecting certain Funders or Politicians with Publishers, the corresponding effect functions are bounded, whereas when demand is lower than half of its maximum value, small increases in demand have an important effect. The slope of the function is higher than that of the effect functions on Academics and Administrators.

*Effect functions of Grant Open Access (GrantOA) on the actors*: Whenever Publishers grant more open access to publications, it is considered in general to be beneficial for all actors except for themselves. There is evidence that toll access leads to more profits in the business of publishing [47], and Publishers therefore sacrifice commercial gains when they grant open access to publications. However, we also wanted to reflect that allowing some OA in their services could benefit Publishers by showing, to a limited extent, that money is not their only concern. This allows them to remain in line with the interests of a wide part of

the market. Additionally, many Publishers benefit from open-access publications because open access is allowed under a certain fee. In a capitalist environment, Politicians face allegations that insisting on OA is equivalent to intervening against market forces, which would be considered harmful in that context. Thus, the effect function increases up until a certain point, and then decreases.

*Effect functions of Demand Open Access (DemandOA) on the actors:* As we said before, we wanted to reflect that in this scenario, OA is beneficial for most members of society, except for Publishers. Therefore, political mandates for all scientific results to be published with OA will cause exponential benefits for Academics, Administrators and Funders. Only Politicians themselves would be subjected to the pro-capital allegations we referred to above, which could somehow bound their actual benefits. Of course, Publishers will see losses, except when there is minimum compliance with those mandates.

2.    Scenarios 1–4.

Below, the changes made in the Scenario 0 model to obtain each new scenario (e.g., variations of effect functions) are described. More details about Scenario 1 are also provided, highlighting the differences with Scenario 0.

- Scenario 1. This scenario was set in the SocLab model by cancelling the impact of all resources for Politicians and Funders, that is, by setting the interest of these actors in all resources to zero. As mentioned above, this represents a situation that is common in many regions around the world, including in Latin America, Africa, Asia, and some developed countries, where Politicians and Funders are scarcely involved in the game; that is, they are inactive with respect to Scenario 0. Their only active role is a function for the solidarity of Politicians with the rest of the actors. Politicians and Funders are too "timid", cautious or disengaged to assume an active role in favor of open access. This could be due to the weakness of the national states and their institutions, the vested interests of some officials against open access, and even a lack of clarity and understanding of the usefulness of open access for national and public benefit.
- Scenario 2. In this scenario, Publishers are less permissive with regard to open access than in Scenario 0. The effect function of GrantOA on Publishers is a straight line with its maximum value on the leftmost side (GrantOA = −10, corresponding to effect = 10; see the graph in the middle of Figure A1). This and the next scenario aim to investigate how a more active position of Publishers targeting their own benefit affects the game and open access. In Scenario 0, Publishers are somewhat permissive with respect to open access; the best impact of the effect function they have on themselves does not occur when support to toll access is at its maximum (see the left side of the effect function of GrantOA on Publishers in Table A1, or the graph on the left side of Figure A1), but rather when it is about a quarter to the right of the left-most point. By contrast, in Scenario 2, Publishers are less permissive: the effect function of GrantOA on Publishers is a straight line, with its maximum value being the leftmost value (GrantOA = −10, corresponding to effect = 10; see the graph in the middle of Figure A1).
- Scenario 3. Publishers are even less permissive than in Scenario 2. The effect function of resource GrantOA on Publishers has a stronger negative slope than in Scenario 2 (see Figure A1). This scenario is similar to Scenario 2, but assumes that Publishers are even less permissive: The effect function of the resource GrantOA on Publishers has a stronger negative slope than in Scenario 2 (see the right side of Figure A1).
- Scenario 4. Funders direct a considerable proportion of their positive solidarity towards Academics, specifically, 0.4 of the total (1). We investigate how the higher awareness on the part of Funders of the importance of open access compared to Scenario 0 promotes open access. This solidarity indicates that the satisfaction of Funders is composed of the pondered sum of two capabilities, theirs and that of Academics, which means that Funders act against actions disfavoring Academics' satisfaction. Funders will manage their resources in favor of their own satisfaction and that of Academics. To understand one of the implications of this, a comparison of the direct

impact of the effect function they exert on Funders can be performed between Scenario 0 and this scenario. In contrast to Scenario 0, in this scenario (Scenario 4), the effect of the resource on Academics' satisfaction needs to be considered, as this impacts Funders' satisfaction, given the solidarity of Funders with Academics. In Scenario 0, to increase Funders' satisfaction, only the impact of the resource they control on themselves is important; thus, it is in Funders' interest to promote open access only when the resource they control is below the middle value (0). The slope of this effect function on themselves is above 0 only when the resource is below 0, as shown in row 3, column 3 of Table A1. However, in this scenario (Scenario 4), Funders will also react in favor of open access, increasing the values of the resource they control when it is above 0 (i.e., in the range [0, 10]), given that this is in the interest of Academics. This encourages Academics' satisfaction, and therefore Funders' satisfaction, as the effect function of the resource controlled by Funders has a positive slope over Academics in the range [0, 10] (see row 2, column 1 of Table A1).

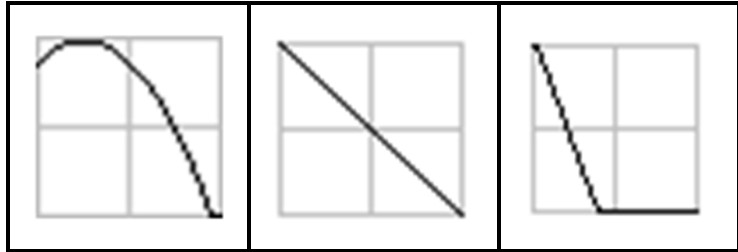

**Figure A1.** Effect functions of the GrantOA resource on Publishers in Scenario 0 (left side), Scenario 2 (middle) and Scenario 3 (right side).

## Notes

1   For instance, the OSTP Nelson memo's goal is declared as: "Ensuring Free, Immediate, and Equitable Access to Federally Funded Research" (Ensuring Free, Immediate, and Equitable Access to Federally Funded Research: Memorandum for the heads of executive departments and agencies, p. 1), with its first suggestion for federal agencies being as follows: "Update their public access policies as soon as possible, and no later than 31 December 2025, to make publications and their supporting data resulting from federally funded research publicly accessible without an embargo on their free and public release; 2. Establish transparent procedures that ensure scientific and research integrity is maintained in public access policies" (ibid, p. 1).

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
