# Peer review of "Simulating and Contrasting the Game of Open Access in Diverse Cultural Contexts: A Social Simulation Model"

_publications, doi:10.3390/publications11030040_

Round 1
Reviewer 1 Report (Previous Reviewer 2)
I appreciate the effort the authors have made to improve this article. That said, I remain concerned that this is just an academic exercise, with little likelihood of accurately describing or predicting how the scholarly publishing enterprise will change with open access policies.
Author Response
We are very grateful to your earlier comments. We have revised the English in the introduction and the conclusion to leave as clear as we can that this is indeed a social simulation exercise, in which we have made a set of assumptions and beliefs explicit, in difference scenarios, to deduce their possible consequences in the context of Open Access activism. We believe that this exercise will be valuable in understanding dynamics in situations where complete information is not available for prediction.
Reviewer 2 Report (New Reviewer)
Review of “Simulating and contrasting the Game of Open Access in diverse cultural contexts”
1. Title could be clearer, for example, specifying the “game of interests” concept or adding something like “a SocLab experiment” or “a social simulation model”.
2. Abstract is fine.
3. In-text citations [square brackets] should be better used, not replacing the name of authors. Also, lines 36 and next should be rewritten to include only numbers.
4. Introduction is ok, although some research questions might be added.
5. Section 2 is fine, and I agree with the options made.
6. Section 3 is ok.
7. Section 4 is very interesting, and the scenarios very well explained.
8. Section 5 presents very well the simulations made.
9. Discussion and Conclusions are also ok.
10. I have only one general commentary. The OA movement was born among the scientists, claiming the control of the scholarly publication system. In fact, only very good reasons would allow someone to offer his/her own work to a commercial publisher, who is going to sell it back to the same researchers/scientific institutions. I would like to read more discussion on the theme of assessment/evaluation (2 words that do not appear in the draft) of researchers/scientific institutions. Assessment is a key issue in the OA discussion, as the citation (and therefore the Impact Factor and other measures) continue to be the normal currency of this social system.
This papers deserves a large revision of the English language.
Author Response
Thank you for your very helpful review. These are the answers to those items that required actions:
1.- The title has been change by adding the expression: a Social Simulation Model.
3.- We tried to adjust the text to combine authors and references in a clearer way. Also, we line 36 and the sourrounding were simplified to leave only a compact reference.
4.- We tried to clarify why we focussed on the declared questions.
10.- We have extended section 2 (from lines 184 to 200) to address this concern about assesment and evaluation of research. We also revisited the whole document to improve the English expressions (with the help of a language advisor) and left some of them marked in blue for easier referencing.
Round 2
Reviewer 2 Report (New Reviewer)
Thanks for the improvements made.
This manuscript is a resubmission of an earlier submission. The following is a list of the peer review reports and author responses from that submission.
Round 1
Reviewer 1 Report
The authors introduce an agent-based model representing the one-shot interactions of the different types of agents participating in the publication market. The cumulative effects of the actions of the interacting agents are assumed to determine the extent to which the publication market is favorable to open access publication practices. The authors assess how different configurations of players' actions promote or hinder open access publications.
The idea to use agent-based models to study the interactive effects of agents' choices in publication markets is conceptually and methodologically sound and fairly innovative, and may offer new insights into the fundamental features of this complex market. Unfortunately, the actual implementation of the model, the analysis of results, and their presentation in this manuscript have major theoretical, technical and methodological shortcomings. The authors should be invited to address the following issues:
1. One-shot interaction model for an obviously sequential problem: some of the chosen players types have significantly higher influence on the publication market and by acting first effectively constrain, either by legal or resource management tools, the action choices of other players. For example, state-level funding bodies may impose legal or resource constraints on the actions of funding recipients, thus constraining their agency in this game. In addition, some players already observe the action choices of the players who act before them, and this should have influence on their assessment of actions. A one-shot model that represents the players as acting simultaneously and with static beliefs about each other's choices fails to capture the salient sequential features of the publication market that are crucially important for the analysis presented in this paper. The authors should consider a Stackelberg leadership model as a more appropriate representation of the publication market.
2. Lack of methodological clarity and rigor: the authors fail to provide a clear and coherent outline of the criteria by which they add simplifying assumptions into their model. The authors fail to provide any logical-argumentative and/or empirical (i.e. non-intuition-based) justification for their choices of the basic parameters of the model, such as "solidarities" (table 4) and "stakes" (table 5) As a result, it is unclear what set of scientific criteria determines the player, action and scenario choices in their model, and the results obtained from simulations based on arbitrary parameter values are of questionable scientific significance. As far as I can see, the authors have two options: either to study a wide variety of possible combinations of parameter values, or provide a sound scientific justification for their choices of parameter values outlined in section 4.
3. Lack of methodological transparency: the authors do not provide a thorough formal characterization of their agent-based model. For example, authors provide a vague formal description of the "capability", "satisfaction" and "influence" functions. It is suggested that the elements of those functions, such as "solidarity", are also functions. Yet they never are formally introduced and so it is impossible to determine the mathematical properties of those functions, as well as the properties of the functions that depend on them. It is virtually impossible to fully grasp the structure of the model and identify the role that each of the chosen parameters plays in producing the final results. Consequently, the current presentation of the model makes it almost impossible to evaluate its conceptual and mathematical soundness.
4. Inadequate analysis of the basic game: the authors do not provide a clear characterization of the structure of the basic game that gets generated by their functions and, contrary to their claims, fail to provide a proper equilibrium analysis of the game. The authors claim to provide a configuration of resource values that establishes a Nash equilibrium of the game. At the very minimum, a proper equilibrium analysis of the game should identify every pure (i.e. non-randomized) combination of parameter values that establishes a Nash equilibrium (in pure actions). It is unclear whether a Nash equilibrium identified by the authors is unique. If it is unique, then the dynamic agent-based analysis has virtually no added value as a dynamic system based on the action revision rule informally described by authors on p. 6 should always converge to that unique equilibrium state.
5. Anomalous results: The dynamic strategy revision rule suggested by authors on p. 6 is, essentially, a short-sighted best reply rule, and so the players should always converge to a stationary state that is a Nash equilibrium. Yet the results provided by the authors seem to show that the dynamic system does not converge to the Nash equilibrium of the game. If the Nash equilibrium is unique, then the results are anomalous under most dynamic revision rules and require rigorous analysis and explanation. If the Nash equilibrium is not unique, then the analysis should be supported by the proper equilibrium analysis of the game that should identify the obtained stationary state as another Nash equilibrium.
6. Inadequate analysis of the dynamic system. Nash equilibria are stationary states under most types of evolutionary dynamics. However, authors' solution criterion - a convergence of a population to a state where no player seeks to adjust their action - is methodologically inadequate as stationary states may be unable to resist even minor exogenous shocks. This is the primary reason why evolutionary game theory prioritizes stronger solution concepts than Nash equilibrium, such as the ESS.
7. Presentation issues: the presentation of the model and various results is very convoluted and extremely difficult to follow. For example, the crucial elements of the model are explained in different parts of the text, some of them are not explained at all. The authors should consider providing a thorough and detailed description of all the relevant parameters of the model in one particular section of the paper.
Reviewer 2 Report
When agreeing to review this manuscript I expected to find a novel and insightful analysis of the practice of open access publishing. The approach is indeed novel -- applying game theory to the primary actors in the scholarly publishing arena: funders, authors, publishers, administrators, and policy makers (a better term than politicians). So while the approach is indeed novel I cannot say the same thing for its being insightful.
The problem is that the authors are forced to make many simplifying assumptions about the roles and responsibilities of the "players" in the game. There are many many subtleties when it comes to scholarly publishing. For example, authors can have many nuanced views of open access: they want it for the materials they consume, but not necessarily for the articles they publish. Administrators are largely interested in the impact of the articles published by the faculty at their universities, typically measured by citations. Here again, though, the landscape is complicated. Open access might allow more readers, but publication in a high impact journal lacking open access could be more important. For both authors and administrators, the potentially higher cost of open access article processing fees could be a negative influencer.
Funders, of course, want to see the results of the research they support get published. Increasingly we are seeing federal policies that require publication in open access journals. (Interesting to note that the authors do not mention the August 2022 policy on open access released by the US Office of Science and Technology Policy, or the European Plan S, both of which have major implications for open access publishing and data sharing.). But we are also seeing a plethora of new predatory journals that promise quick publication and open access, but in fact provide little if any peer review or quality assessment. None of these aspects are represented in the analysis in this paper.
And not all publishers are the same, simply motivated by profit. There are many no-for-profit publishers, i.e., particularly those associated with professional societies, for whom serving their research communities is a higher priority than financial gain.
Thus, overall, I find the game theory analysis to be academically interesting but of little overall value given the oversimplification of the realities of the academic publishing world. The conclusion that open access publishing will be catalyzed first in Europe and the US is actually pretty obvious without the analysis done here. Plan S in Europe and the OSTP "Nelson memo" of August 2022 will undoubtedly lead to a new equilibrium in the academic publishing arena, and the major for-profit publishers will find income streams other than subscriptions to maintain their financial stability. Funders will have to permit grantees to pay potentially higher APCs, but these increases to grant budgets will, in time, be offset by decreased expenses to university libraries for journal subscriptions (typically covered in overhead rates).